# Mitigating Exploitation Bias in Learning to Rank with an Uncertainty-aware Empirical Bayes Approach

## Abstract

Ranking is at the core of many artificial intelligence (AI) applications, including search engines, recommender systems, etc. Modern ranking systems are often constructed with learning-to-rank (LTR) models built from user behavior signals. While previous studies have demonstrated the effectiveness of using user behavior signals (e.g., clicks) as both features and labels of LTR algorithms, we argue that existing LTR algorithms that indiscriminately treat behavior and non-behavior signals in input features could lead to suboptimal performance in practice. Because user behavior signals often have strong correlations with the ranking objective and can only be collected on items that have already been shown to users, directly using behavior signals in LTR could create an exploitation bias that hurts the system performance in the long run.

To address the exploitation bias, we propose an uncertainty-aware empirical Bayes based ranking algorithm, referred to as EBRank. Specifically, EBRank uses a sole non-behavior feature-based prior model to get a prior estimation of relevance. In the dynamic training and serving of ranking systems, EBRank uses the observed user behaviors to update posterior relevance estimation instead of concatenating behaviors as features in ranking models. Besides, EBRank additionally applies an uncertainty-aware exploration strategy to explore actively and collect user behaviors for empirical Bayesian modeling. Experiments on three public datasets show that EBRank is effective, practical and significantly outperforms state-of-the-art ranking algorithms.

## Keywords

Learning to rank, Behavior feature, Exploitation bias

## 1 Introduction

Ranking techniques have been extensively studied and used in modern Information Retrieval (IR) systems such as search engines, recommender systems, etc. Among different ranking techniques, learning to rank (LTR), which relies on building machine learning (ML) models to rank items, is one of the most popular ranking frameworks [28]. In particular, industrial LTR systems are usually constructed with user behavior feedback/signals since user behaviors (e.g., click, purchase) are cheap to get and directly indicate results' relevance from the user's perspective [50]. For example, previous studies [6, 23, 50] have shown that, instead of using expensive relevance annotations from experts, effective LTR models can be learned directly from training labels constructed with user clicks. Besides using clicks as training labels, many industrial IR systems have also considered features extracted from user clicks for their LTR models. For example, ranking features extracted from clicks

are used in search engines like Yahoo and Bing [9, 37]. Agichtein et al. [4] has also shown that by incorporating such ranking features in ranking systems, the performance of competitive web search ranking algorithms can be improved by as much as 31%.

However, without proper treatment, LTR with user behaviors can also damage ranking quality in the long term [24, 25, 54]. Specifically, user behavior signals usually have high correlations with item relevance labels since behavior signals are direct and strong indicators of relevance from the user's perspective. Such high correlation can easily make input features built from user behavior signals, referred to as the behavior features, overwhelm other features in training, dominate model outputs, and be over-exploited in inference. Such an over-exploitation phenomenon would hurt practical ranking systems when user behavior signals are unevenly collected on different candidate items [18, 25]. For example, we can only collect user clicks on items already presented to users. Items that lack historical click data, including new items that have not yet been presented to users, would be at a disadvantage. The disadvantage, referred to as the exploitation bias [54], can be more severe when we use user clicks/behaviors as both labels and features, which is a common practice in real-world LTR systems [9, 17, 18, 37, 54]. One similar concept is selection bias [32]. Selection bias usually refers to the bias that occurs when user clicks are used as training labels. In contrast, exploitation bias goes one step further and considers the bias that arises in a more realistic scenario where user clicks/behaviors are also used as ranking features.

In this paper, we address the above exploitation bias with an uncertainty-aware empirical Bayesian based algorithm, EBRank. Specifically, we consider a general application scenario where a ranking system is built with user behavior signals (i.e., clicks in this paper) in both its input and objective functions. We show that, without differentiating the treatment of behavior signals and non-behavior signals in input features, existing LTR algorithms could suffer severely from exploitation bias. By differentiating behavior signals and non-behavior signals, the proposed algorithm, EBRank, uses a sole non-behavior feature based prior model to give a prior relevance estimation. With more behavior data collected from the online serving process of a ranking system, EBRank gradually updates its posterior relevance estimation to give a more accurate relevance estimation. Besides, we also proposed a theoretically principled exploration algorithm that joins the optimization of ranking performance with the minimization of model uncertainty. Experiments on three public datasets show that our proposed algorithm can effectively alleviate exploitation bias and deliver superior ranking performance compared to state-of-the-art ranking algorithms.

## 2 Related Work

***Ranking exploitation with behavior features.*** As an important relevance indicator, user behavior signals have been important components for constructing modern IR systems [37]. [4, 29] showed that incorporating user behavior data as features can significantly

*Conference acronym 'XX, June 03–05, 2018, Woodstock, NY*
© 2018 Association for Computing Machinery.
ACM ISBN 978-1-4503-XXXX-X/18/06...$15.00
https://doi.org/XXXXXXX.XXXXXXX

**Table 1: A summary of notations.**

| | |
|---|---|
| $d, q, D(q)$ | For a query $q$, $D(q)$ is the set of candidates items. $d \in D(q)$ is an item to rank. |
| $e, r, c$ | All are binary random variables indicating whether an item $d$ is examined ($e = 1$), is relevant ($r = 1$) and clicked ($c = 1$) by a user respectively. |
| $R, \rho, E, \pi, n$ | $R = P(r = 1)$, is the probability an item $d$ perceived as relevant. $\rho = P(e = 1)$ is the examination probability. $\pi$ is a ranklist. $E$ is an item's accumulated examination probability. $n$ is the number of times item $d$ has been presented to users (see Eq.9). |
| $k_s, k_c$ | Users will stop examining items lower than rank $k_s$ due to selection bias (see Eq. (3). $k_c$ is the cutoff prefix to evaluate Cum-NDCG and $k_c \leq k_s$. |
| $x^b, x^{nb}$ | $x^b$ denotes ranking features derived from user feedback behavior, while $x^{nb}$ denotes ranking features derived from non-behavior features. |

improve the ranking performance of top results. However, incorporating behavior features without proper treatments could hurt the effectiveness of LTR systems by amplifying the problem of over-exploitation and over-fitting, i.e., exploitation bias [54]. Kveton et al. [24], Li et al. [25], Oosterhuis and de Rijke [34] discussed the generation problems and cold-start problems. Some strategies were proposed to overcome the exploitation bias by predicting behavior features with non-behavior features [18, 19] or by actively collecting user behavior for new items [25, 54] via exploration.

***Unbiased/Online Learning to Rank***. Using biased and noisy user clicks as training labels for LTR has been extensively studied in the last decades [7, 35, 45]. Among different unbiased LTR methods, online LTR chooses to actively remove the bias with intervention based on bandit learning [47, 58] or stochastic ranking sampling [31]. In contrast, offline LTR methods usually train LTR models with offline click logs based on techniques such as counterfactual learning [2, 5, 23, 53]. Existing unbiased LTR methods effectively remove bias when clicks are treated as labels. However, in this work, we investigate and show that existing unbiased LTR methods still suffer the exploitation bias.

***Uncertainty in ranking***. One of the first studies of uncertainty in IR is Zhu et al. [57], where the variance of a probabilistic language model was treated as a risk-based factor to improve retrieval performance. Recently, uncertainty estimation techniques for deep learning models have also been introduced into the studies of neural IR models [12, 36, 49]. Uncertainty quantification is an important IR community for many downstream tasks. For example, [11, 20, 22, 26, 54] proposed uncertainty-aware exploration ranking algorithms. Existing studies have also shown that uncertainty can help improve query performance prediction [39], query cutoff prediction [14, 27], and ranking fairness [55].

## 3 Background

In this section, we introduce some preliminary knowledge.

**The Workflow of Ranking Services**. In Figure 1, we use web search as an example to introduce the workflow of ranking service in detail, but the method we propose in this paper can also be extended to the recommendation or other ranking scenarios. At time step $t$, a user issues a query $q_t$. Corresponding to this query, there exist candidate items which include old items and new items introduced at time step $t$. These items are represented as features, which

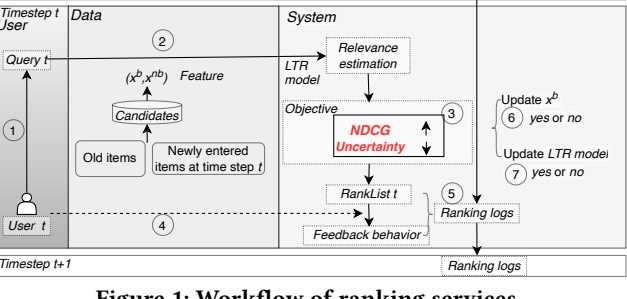

**Figure 1: Workflow of ranking services.**

include behavior signals $x^b$ and non-behavior features $x^{nb}$ (a detailed discussion about features is given in the following paragraph). Based on the features, an LTR model will predict the relevance of each candidate item, and the ranking optimization methods will generate the ranked list by optimizing some ranking objectives. Different ranking objectives can be adopted here, such as maximizing NDCG (see Eq. (6)). After examining the ranked list, the user will provide behavior feedback, such as clicks. The rank list and user feedback behavior will be appended to the ranking logs. We need to decide whether to update the LTR model and behavior features or not. In practice, such updates are usually conducted periodically since real-time updating ranking systems are often not preferable.

**Features.** In this paper, LTR features are categorized into two groups based on Qin et al. [37]. The first group is non-behavior features, denoted as $x^{nb}$, which show items' quality and the degree of matching between item and query. Example $x^{nb}$ can be BM25, query length, tf-idf, features from pre-trained (large) language models, etc. Non-behavior features $x^{nb}$ are usually stable and static. The second group is behavior features, denoted as $x^b$, which are usually derived from user behaviors, e.g., click-through-rate, dwelling time, click, etc. $x^b$ are direct and strong indicators of relevance from the user's perspective since $x^b$ are collected directly from the user themselves. Unlike $x^{nb}$, $x^b$ are dynamically changing and constantly updated. In this paper, we only focus on one type of user behavior data, i.e., clicks. Extending our work to other types of user behaviors is straightforward, and we leave them for future studies.

**Partial and Biased User Behavior.** Although user behavior is commonly used in LTR, user behavior is usually biased and partial. Specifically, a user will provide meaningful feedback click ($c = 1$) only when a user examines ($e = 1$) the item, i.e.,

$$c = \begin{cases} r, & \text{if } e = 1 \\ 0, & \text{otherwise} \end{cases} \quad (1)$$

where $r$ indicates if a user would find an item $d$ as relevant. Here $c, r, e$ are random binary variables. A detailed summary of notations is in Table 1. Following [6, 50], we model users' click behavior as,

$$P(c = 1) = P(r = 1)P(e = 1) \quad (2)$$

For items ranked on different positions in the user interface, the examination probability $P(e = 1)$ are usually different. In this paper, we model examination differences with the two most important biases, i.e., positional bias and selection bias. The **position bias** [13] assumes the examination probability of an item $d$ in a ranklist $\pi$ solely relies on the rank (position), i.e., $rank(d|\pi)$, and we model it with $\rho_{rank(d|\pi)} = P(e = 1|d, \pi)$ to simplify notation. The **selection**

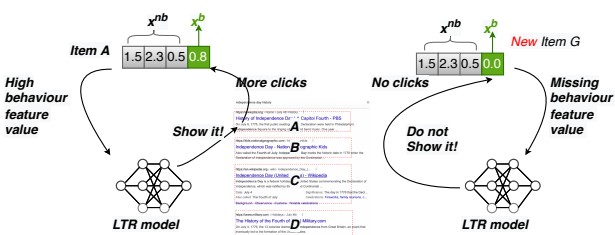

**Figure 2: Toy example to show the exploitation bias. Old item A and new item G have the same non-behavior ($x^{nb}$) features, while item G's behavior features ($x^b$) is 0 as it has not been shown to users before. Item G will be discriminated against if the LTR model over-exploits and heavily relies on $x^b$.**

bias [32, 33] exists when not all of the items are selected to be shown to users, or some lists are so long that no user will examine the entire lists. The selection bias is modeled as,

$$\rho_{rank(d|\pi)} = 0 \quad \text{if} \quad rank(d|\pi) > k_s \tag{3}$$

where $k_s$ is the lowest rank that will be examined.

**Exploitation Bias.** In Figure 2, we give a toy example to illustrate the exploitation bias. The exploitation bias usually happens because behavior features in LTR models will overwhelm other features since behavior features are strong indicators of relevance and highly correlated with training labels. In this example in Figure 2, due to the overwhelming importance of behavior features, newly introduced item G will be discriminated by the LTR model and ranked low when its user behavior information is missing.

**Ranking utility.** In this paper, ranking utility is evaluated by measuring a ranking system's ability to put relevant items on top ranks effectively. Firstly, the relevance for a $(d, q)$ pair is

$$R(d, q) = p(r = 1 | d, q) \tag{4}$$

is defined as the probability of an item $d$ being considered relevant to a given query $q$. One widely-used ranking utility measurement is $DCG$ [21]. For a ranked list $\pi$ corresponding to a query $q$, we have

$$DCG@k_c(\pi) = \sum_{i=1}^{k_c} R(\pi[i], q)\lambda_i \tag{5}$$

where $\pi[i]$ indicates the $i^{th}$ ranked item in the ranked list $\pi$; $R(\pi[i], q)$ indicates item $\pi[i]$'s relevance to query $q$; $\lambda_i$ indicates the weight we put on $i^{th}$ rank; cutoff $k_c$ indicates the top $k_c$ ranks we evaluate. In this paper, when cutoff $k_c$ is not specified, there is no cutoff. $\lambda_i$ usually decreases as rank $i$ increases since top ranks are usually more important. For example, $\lambda_i$ is sometimes set to $\frac{1}{\log_2(i+1)}$. In this paper, we follow [43] to choose the $i^{th}$ rank's examining probability $\rho_i$ as $\lambda_i$. Then, we can get normalized-$DCG$ ($NDCG \in [0, 1]$) by normalizing $DCG@k_c(\pi)$,

$$NDCG@k_c(\pi) = \frac{DCG@k_c(\pi)}{DCG@k_c(\pi^*)} \tag{6}$$

where $\pi^*$ is the ideal ranked list constructed by arranging items according to their true relevance. Furthermore, we could define discounted Cumulative NDCG (Cum-NDCG) as ranking effectiveness,

$$Cum\text{-}NDCG@k_c = \sum_{\tau=1}^{t} \gamma^{t-\tau} NDCG@k_c(\pi_\tau) \tag{7}$$

where $0 \leq \gamma \leq 1$ is the discounted factor, $t$ is the current time step. Note that $\gamma$ is a constant number for evaluating ranking performance [48]. Compared to NDCG, Cum-NDCG can better evaluate ranking effectiveness for online ranking services [41].

**Uncertainty in relevance estimation.** In real-world applications, the true relevance $R$ is usually unavailable. Relevance estimation, denoted as $\hat{R}$, is usually needed for ranking optimization. However, $\hat{R}$ usually contains uncertainty (variance), denoted as $Var[\hat{R}]$. Furthermore, we introduce the query-level uncertainty,

$$U(q) = \sum_{d \in D(q)} Var[\hat{R}(d, q)] \tag{8}$$

which will be used to guide ranking exploration. We leave more advanced query-level uncertainty formulations for future study.

## 4 Proposed Method

In this section, we propose an uncertainty-aware Empirical Bayes based learning to rank algorithm, EBRank, which can effectively mitigate exploitation bias. We formally introduce the proposed algorithm EBRank In Sec .4.1. And we dive into the theoretical parts of EBRank, which include ranking objectives, Empirical Bayes modeling, and uncertainty reduction in Sec. 4.2, 4.3 and 4.4, repectively.

### 4.1 The proposed algorithm: EBRank

In this section, we give a big picture of the proposed Empirical Bayesian Ranking (EBRank) in Algorithm 1. Prior to serving users, we initialize a list, $\mathcal{H}$, to store ranking logs. At each time step, we append four elements, $[u_t, q_t, \pi_t, c_t]$, to $\mathcal{H}$. Here, $u_t, q_t, \pi_t, c_t$ are the user, the query, the presented ranklist, and the user clicks at time step $t$. If initial ranking logs exist prior to using EBRank, we will also append them to $\mathcal{H}$. Besides ranking logs, we initialize a model $f_\theta$, referred to as the prior model, parameterized by $\theta$, which takes non-behavior features as input. $f_\theta$ can be any trainable parameterized model, such as a vanilla neural network, tree-based model, etc. When EBRank begins to serve users ($t > 0$), new candidates will be constantly appended to each query's candidates set, $D(q_t)$. Then, for each item-query pair $(d, q_t)$, we construct an auxiliary list $\mathcal{A}(d, q_t) = [\alpha, \beta, n, C, E]$, where

$$[\alpha, \beta] = f_\theta(x^{nb}(d, q_t))$$
$$n = \sum_{\tau \in T(t,d,q_t)} 1$$
$$C = \sum_{\tau \in T(t,d,q_t)} \frac{c(d|\pi_\tau)}{\rho_{rank(d|\pi_\tau)}} \tag{9}$$
$$E = \sum_{\tau \in T(t,d,q_t)} \rho_{rank(d|\pi_\tau)}$$

$$T(t, d, q_t) = \left[ \tau \quad \text{if} \quad \mathbb{I}(d \in \pi_\tau)\mathbb{I}(q_\tau = q_t) \quad \text{for} \quad 0 \leq \tau < t \right]$$

The non-behavior features $x^{nb}(d, q_t)$ are usually static and the prior model, $f_\theta$, directly takes $x^{nb}(d, q_t)$ as the input and exports two numbers, i.e., $[\alpha, \beta]$, for each item-query pair. $T(t, d, q_t)$ is a subset of historical time steps when item $d$ is presented in query $q_t$'s ranklists in the past and $\mathbb{I}$ is the indicator function. $n$ is the number of times that item $d$ has been presented for query $q_t$ by time step $t - 1$. $c(d|\pi_\tau)$ indicates whether item $d$ is clicked or not. $C$ is the sum of weighted clicks on item $d$, and the weight is its examination

probability, $E$ is the item's exposure which is the accumulation of examination probability.

*4.1.1 Relevance estimation.* Based on $\mathcal{A}(d, q_t)$, firstly, we estimate item's relevance $\widehat{R}(d, q_t)$ as,

$$\widehat{R}(d, q_t) = \frac{C + \alpha}{n + \alpha + \beta} \tag{10}$$

which is based on our empirical Bayes modeling in Sec. 4.2. The relevance estimation $\widehat{R}(d, q_t)$ is a blending between $\frac{C}{n}$ and $\frac{\alpha}{\alpha+\beta}$. When $n > 0$, $\frac{C}{n}$ is an unbiased estimation of true relevance $R(d, q_t)$ (see Eq. (22)). Relevance estimation $\frac{C}{n}$ is limited as it requires $n > 0$, i.e., item $d$ has been selected for query $q_t$ before. Without this limitation, $\frac{\alpha}{\alpha+\beta}$, referred to as prior relevance estimation, is solely based on non-behavior features. Similar to $\frac{C}{n}$, $\frac{\alpha}{\alpha+\beta}$ also theoretically approximates the true relevance $R(d, q_t)$ given an perfectly optimized prior model $f_\theta$ (more details is in later sections).

With the blending of the two parts, $\widehat{R}$ can overcome exploitation bias by nature since it will rely more on $\frac{\alpha}{\alpha+\beta}$ when $n$ and $C$ are small, i.e., new items. $\widehat{R}$ gradually relies more on $\frac{C}{n}$ when $n$ and $C$ increase and start to dominate, i.e., more user behaviors are observed.

*4.1.2 Ranking exploration & construction.* Besides relevance, we introduce an exploration score for item $d$,

$$MC(d, q_t) = \frac{\widehat{R}(d, q_t)}{\left(E + \alpha + \beta\right)^2} \tag{11}$$

which boosts candidates that have a higher estimated relevance $\widehat{R}(d, q_t)$ but a lower exposure $E$. The exploration score helps to gain the greatest certainty at the current time step according to our ranking uncertainty analysis in Sec. 4.4, With the relevance estimation and the exploration score, we construct ranklist $\pi_t$ by sorting $\widehat{R}(d, q_t) + \epsilon MC(d, q_t)$ in descending order, i.e.,

$$\pi_t = \arg_{sort\text{-}k} \left( \widehat{R}(d, q_t) + \epsilon MC(d, q_t) | \forall d \in D(q_t) \right) \tag{12}$$

where a possible cutoff $k$ might exist to show the top results only. $\epsilon$ is a hyper-parameter to balance the two parts. The ranklist construction is based on the ranking objective proposed in Sec. 4.2.

*4.1.3 Prior model optimization.* The parameters $\theta$ in $f_\theta$ will be periodically updated with the following loss,

$$l(d, q) = \mathbb{I}_{n>0} \left( \log B(\alpha, \beta) - \log B(C + \alpha, n - C + \beta) \right) \tag{13a}$$

$$\mathcal{L} = \sum_{q \in Q} \sum_{d \in D(q)} l(d, q) \tag{13b}$$

where $[\alpha, \beta, n, C, E] \in \mathcal{A}(d, q)$, $B$ denotes the beta function. Note that $\alpha, \beta$ are $f_\theta$'s outputs. The loss is based on our Empirical Bayes modeling in Sec. 4.3. The above loss to train $f_\theta$ is based on items that have been presented to users before ($n > 0$) since only those items could possibly have user clicks, which is the best we can do. According to our empirical results, $f_\theta(x^{nb})$ generalizes well for items not presented before, i.e., $n = 0$. If there exist some initial ranking logs, $f_\theta$ can also be trained based on them. More analysis of the loss can be found in Sec. 4.3.3.

## 4.2 Uncertainty-Aware ranking optimization

In this section, we explain why Eq. 12 is optimal for constructing ranklists.

---

**Algorithm 1:** EBRank

1   $t \leftarrow 0$;
2   $\mathcal{H} = [u_t, q_t, \pi_t, c_t] \; \forall t \in history$;
3   Initialize the trainable parameters $\theta$ for prior model $f_\theta$;
4   Initialize hyper-parameter $\epsilon$;
5   **while** *True* **do**
6     $t \leftarrow t + 1$;
7     User $u_t$ issues a query $q_t$;
8     **if** *New candidates introduced* **then**
9       $D_{q_t}$.append(new candidates)
10    $\forall d \in D_{q_t}$, construct $\mathcal{A}(d, q_t)$ via Eq. (9);
11    With $\mathcal{A}(d, q_t)$, get $\widehat{R}(d, q_t)$ via Eq. (10);
12    With $\mathcal{A}(d, q_t)$ & $\widehat{R}(d, q_t)$, get $MC(d, q_t)$ via Eq. (26);
13    With $\widehat{R}(d, q_t)$ & $MC(d, q_t)$, get $\pi_t$ via Eq. (12);
14    Present $\pi_t$ to User $u_t$ and collect user's clicks $c_t$;
15    $\mathcal{H}.append([u_t, q_t, \pi_t, c_t])$;
16    **if** *Prior-model-update* **then**
17       Train $f_\theta$ with loss $\mathcal{L}$ in Eq. (13)

---

*4.2.1 The uncertainty-aware ranking objective.* As shown in Figure 1, at time step $t$, a user issues a query $q_t$, and we propose the following uncertainty-aware ranking objective to optimize ranklist $\pi_t$,

$$\max_{\pi_t} \quad Obj = \widehat{DCG}(\pi_t) - \epsilon \Delta U(\pi_t) \tag{14}$$

where $\Delta U(\pi_t) = U(q_t|\pi_t) - U(q_t)$ is the query-level uncertainty increment after presenting $\pi_t$ to the user. $\widehat{DCG}$ is the estimated DCG (see Eq.7) based on the estimated relevance $\widehat{R}$ instead of the unavailable true relevance $R$. $\epsilon$ is the coefficient to balance the two parts. In Eq. (14), our real goal is to maximize the true $DCG$, but we only have the estimated $\widehat{DCG}$ which is calculated based on estimated relevance. Hence, to effectively optimize $DCG$ via the proxy of optimizing $\widehat{DCG}$, we need an accurate $\hat{R}(d, q_t)$, which is why uncertainty is also minimized in Eq. (14). Here we choose to minimize the incremental uncertainty at time step $t$ since only the incremental uncertainty is caused by $\pi_t$.

*4.2.2 Ranking optimization.* To maximize ranking objectives in Eq. (14) and optimize $\pi_t$, we first reformulate $\widehat{DCG}$ in Eq. (5) as,

$$\widehat{DCG}(\pi_t) = \sum_{d \in D(q_t)} \hat{R}(d, q_t) \cdot \rho_{rank(d|\pi_t)} \tag{15}$$

As for $\Delta U(\pi_t)$ in Eq. (14), inspired by [55], we carry out a first-order approximation by considering incremental exposure $\Delta E$,

$$\Delta U(\pi_t) \approx \sum_{d \in D(q_t)} \frac{\partial U(q_t)}{\partial E(d, q_t)} \Delta E(d, q_t)$$
$$= \sum_{d \in D(q_t)} \frac{\partial Var[\hat{R}(d, q_t)]}{\partial E(d, q_t)} \rho_{rank(d|\pi_t)} \tag{16}$$
$$= \sum_{d \in D(q_t)} -MC(d, q_t) \rho_{rank(d|\pi_t)}$$

where $\Delta E(d, q_t)$ is the incremental exposure that item $d$ will get at time $t$, i.e., $\rho_{rank(d|\pi_t)}$, $MC(d, q_t)$ is the gradient of minus variance, denotes **M**arginal **C**ertainty, the speed to gain additional certainty.

Since $\rho_{rank(d|\pi_t)} \in (0, 1)$ is relatively small, the first-order approximation in Eq. (16) should approximate well. In Eq. (16), we assume that $Var[\hat{R}(d_x, q_t)]$ and $E(d_y, q_t)$ are independent for different items $d_x$ and $d_y$, so $\frac{\partial Var[\hat{R}(d_x, q_t)]}{\partial E(d_y, q_t)} = 0$. We will introduce how $Var[\hat{R}(d, q_t)]$ and $E(d, q_t)$ are related in Sec 4.4.

Finally, the ranking objective in Eq. (14) can be rewritten as

$$\max_{\pi_t} \sum_{d \in D(q_t)} \left( \hat{R}(d, q_t) + \epsilon MC(d, q_t) \right) \rho_{rank(d|\pi_t)} \qquad (17)$$

By assuming that $\rho_{rank(d|\pi_t)}$ descends as $rank(d|\pi_t)$ goes lower, the optimal $\pi_t$ should be generated by sorting items according to $\left( \hat{R}(d, q_t) + \epsilon MC(d, q_t) \right)$ in descending order, which validates Eq. (12).

## 4.3 Empirical Bayesian relevance model

In this section, we propose an empirical Bayesian relevance model that leads to relevance estimation Eq. (10) and loss function Eq. (13).

*4.3.1 The observation modelling.* At time step $t$, users issue the query $q_t$. When there is no position bias and the binary relevance judgments $r$ are directly observable, the probability of observation of relevance judgments for a $(d, q_t)$ pair prior to time step $t$ is

$$P(r^*|\bar{R}) = \prod_{\tau \in T(t,d,q_t)} \left( (\bar{R})^{r(d|\pi_\tau)} \times (1 - \bar{R})^{(1-r(d|\pi_\tau))} \right) \qquad (18)$$

where $r^*$ denotes users' relevance judgments. $0 \leq \bar{R} \leq 1$, and $\bar{R}$ is a random variable that denotes our estimated probability of $r = 1$. $r(d|\pi_\tau)$ is the observation of random binary variable $r$ at time $\tau$.

However, user relevance judgments are not observable, and we can only observe user clicks for $\tau < t$, although clicks are biased indicator of relevance according to Eq. (1). In this paper, based on observable clicks, we introduce probability $P(c^*|\bar{R})$ as a proxy for $P(r^*|\bar{R})$,

$$P(c^*|\bar{R}) = \prod_{\tau \in T(t,d,q_t)} \left( (\bar{R})^{\frac{c(d|\pi_\tau)}{\rho_{rank(d|\pi_\tau)}}} \times (1 - \bar{R})^{(1 - \frac{c(d|\pi_\tau)}{\rho_{rank(d|\pi_\tau)}})} \right)$$
$$= (\bar{R})^C \times (1 - (\bar{R}))^{n-C} \qquad (19)$$

where $c^*$ denotes users' clicks. $c(d|\pi_\tau)$ indicates whether item $d$ is clicked or not in ranklist $\pi_\tau$, and $\rho_{rank(d|\pi_\tau)}$ is item $d$'s examining probability in presented ranklist $\pi_\tau$. We use $P(c^*|\bar{R})$ as a proxy of $P(r^*|\bar{R})$ since we noticed that $\log P(c^*|\bar{R})$ is an unbiased estimation of $\log P(r^*|\bar{R})$ [8, 40],

$$\mathbb{E}_e[\log P(c^*|\bar{R})]$$
$$= \sum_{\tau \in T(t,d,q_t)} \left( \frac{\mathbb{E}_e[c(d|\pi_\tau)]}{\rho_{rank(d|\pi_\tau)}} \log(\bar{R}) + (1 - \frac{\mathbb{E}_e[c(d|\pi_\tau)]}{\rho_{rank(d|\pi_\tau)}}) \log(1 - \bar{R}) \right)$$
$$= \log P(r^*|\bar{R})$$

where expectation $\mathbb{E}_e[c(d|\pi_\tau)] = \rho_{rank(d|\pi_\tau)} r(d|\pi_\tau)$ given Eq. (1) and Eq. (2). Although the proof above takes a logarithm and the log-likelihood is unbiased instead of likelihood itself, $P(c^*|\bar{R})$ is still an effective proxy for $P(r^*|\bar{R})$, which is validated by our empirical results. One may note that [8, 40] have similar theoretical analysis here, but our method is fundamentally different from theirs. They use $-\log P(c^*|\bar{R})$ as the final ranking loss function, while we use

$P(c^*|\bar{R})$ as the observation probability, just one part of the many components of our Bayes model.

*4.3.2 The prior & posterior distribution.* Because the formulation $P(c^*|\bar{R})$ is similar to a binomial distribution, we choose the binomial distribution's conjugate prior, the Beta distribution, as the prior distribution, i.e., the prior relevance $\bar{R} \sim Beta(\alpha, \beta)$,

$$P(\bar{R}|\theta) = \frac{\bar{R}^{\alpha-1}(1 - \bar{R})^{\beta-1}}{B(\alpha, \beta)} \qquad (20)$$

where $B$ denotes the beta function.[1] According to the theory of conjugate distribution [38], the posterior distribution of $\bar{R}$ also follows a beta distribution,

$$\bar{R} \sim Beta(C + \alpha, n - C + \beta) \qquad (21)$$

We use the expectation of the above beta distribution, i.e., $\frac{C+\alpha}{n+\alpha+\beta}$, as the posterior relevance estimation $\hat{R}$ (see. Eq. 10). When $n > 0$, $\frac{C}{n}$ is an unbiased estimation of true relevance $R(d, q_t)$)

$$\mathbb{E}_c\left[\frac{C}{n}\right] = \frac{1}{n} \sum_{\tau \in T(t,d,q_t)} \frac{\mathbb{E}_c[c(d|\pi_\tau)]}{\rho_{rank(d|\pi_\tau)}}$$
$$= \frac{1}{n} \sum_{\tau \in T(t,d,q_t)} \frac{\rho_{rank(d|\pi_\tau)} R(d, q_t)}{\rho_{rank(d|\pi_\tau)}} = R(d, q_t) \qquad (22)$$

*4.3.3 Update prior distribution with observations.* Here, we reveal how to get the loss in Eq 13a. We perform the maximum a posterior (MAP) [16] by maximizing the marginal likelihood of the observed data i.e., $P(c^*|\theta)$.

$$P(c^*|\theta) = \int_{\bar{R}} P(c^*|\bar{R}) P(\bar{R}|\theta) d\bar{R}$$
$$= \frac{\int_0^1 \bar{R}^{C+\alpha-1}(1 - \bar{R})^{n-C+\beta-1} d\bar{R}}{B(\alpha, \beta)} \qquad (23)$$
$$= \frac{B(C + \alpha, n - C + \beta)}{B(\alpha, \beta)}$$

Maximizing the above probability is also equivalent to minimizing $-\log P(c^*|\theta)$, i.e., the loss function in Eq 13a.

To investigate what is the optimal $(\alpha, \beta)$ during training, we take the derivative of the loss function in Eq 13a,

$$\frac{\partial l(d, q)}{\partial \alpha} = \psi(\alpha) - \psi(\alpha + \beta) - \left( \psi(C + \alpha) - \psi(n + \alpha + \beta) \right) \qquad (24)$$

where $\psi$ is the Digamma function. Usually, by setting, $\frac{\partial l(d,q)}{\partial \alpha} = 0$, we could know the optimal $(\alpha^*, \beta^*)$. However, to our knowledge, it is difficult to directly get $(\alpha^*, \beta^*)$ from the above equation. Since $\psi(x) \approx \log(x)$ with error $O(\frac{1}{x})$[1], we use log function to substitute $\psi$ in Eq. (24), and set $\frac{\partial l(d,q)}{\partial \alpha} = 0$, and we get,

$$\frac{\alpha^*}{\alpha^* + \beta^*} - \frac{C + \alpha^*}{n + \alpha^* + \beta^*} = 0 \qquad (25)$$

It is straightforward to see that the optimal prior estimation should output $(\alpha^*, \beta^*)$ that satisfies $\frac{\alpha^*}{\alpha^* + \beta^*} = \frac{C}{n}$, where $\frac{C}{n}$ is an unbiased estimation of relevance $\bar{R}$ given Eq. (22) as long as $n > 0$. In Eq. (25), $\alpha^*, \beta^*$ are not unique. To make $\alpha, \beta$ in Eq. (25) have unique values, we fix $\beta$ and only learn $\alpha$ for simplicity. We leave how to get unique $\alpha, \beta$ without fixing one of them to future works.

---

[1]The beta distribution and the beta function are denoted as $Beta$ and $B$ respectively.

## 4.4 Estimation of Marginal Certainty.

In this section, we introduce how to get the exploration score in Eq. 11. As indicated in Eq. (8&16), to get the $MC(d, q_t)$, we first need to get the variance of $\hat{R}(d, q_t)$. For simplicity, we assume that the only random variable in $\hat{R}(d, q_t)$ is $c(d|\pi_\tau)$, where $c(d|\pi_\tau)$ and $\hat{R}(d, q_t)$ are associated via $C$. Firstly, the variance of $c(d|\pi_\tau)$ is

$$Var[c(d|\pi_\tau)] = \mathbb{E}_c[c^2(d|\pi_\tau)] - \mathbb{E}_c^2[c(d|\pi_\tau)]$$
$$= \mathbb{E}_c[c(d|\pi_\tau)] - \mathbb{E}_c^2[c(d|\pi_\tau)]$$
$$< \rho_{rank(d|\pi_\tau)} R(d, q_t)$$

where $c^2(d|\pi_\tau) = c(d|\pi_\tau)$ since $c(d|\pi_\tau)$ is a binary random variable. $\mathbb{E}_c[c(d|\pi_\tau)] = \rho_{rank(d|\pi_\tau)} R(d|\pi_t)$ according to Eq. 2. According to the linearity of expectation, we can get the variance of $\hat{R}(d, q_t)$,

$$Var[\hat{R}(d|q_t)] = \frac{Var[C]}{(n+\alpha+\beta)^2} = \frac{\sum_{\tau \in T(t,d,q_t)} \frac{1}{\rho_{rank(d|\pi_\tau)}^2} Var[c(d|\pi_\tau)]}{(n+\alpha+\beta)^2}$$
$$< \frac{R(d, q_t) \sum_{\tau \in T(t,d,q_t)} \frac{1}{\rho_{rank(d|\pi_\tau)}}}{(n+\alpha+\beta)^2} < \frac{R(d, q_t)\frac{n}{\rho_{min}}}{(n+\alpha+\beta)^2}$$
$$< \frac{R(d, q_t)1/\rho_{min}}{(n+\alpha+\beta)} \doteq \frac{R(d, q_t)}{E+\alpha+\beta}$$

where we assume there exists the smallest examining probability $\rho_{min}$ for presented item $d$. In the last step, we ignore the constant factor, i.e., $1/\rho_{min}$. We use the above upper bound as an approximation of variance, which works well according to our empirical results. Given Eq. (16), we can get $MC(d, q_t)$ by taking derivative of $(-Var[\hat{R}])$ with respect to $E$,

$$MC(d, q_t) = \frac{\partial(-Var[\hat{R}(d|q_t)])}{\partial E} \approx \frac{\hat{R}(d, q_t)}{(E+\alpha+\beta)^2} \quad (26)$$

where we use $\hat{R}$ to substitute relevance $R$ since $R$ is unavailable.

## 5 Experiments

In this section, we evaluate the effectiveness of the proposed method with semi-simulation experiments on public LTR datasets. To facilitate reproducibility, we release our code [2]

### 5.1 Experimental setup

*5.1.1 Dataset.* In the semi-simulation experiments, we will adopt three publicly available LTR datasets, i.e., MQ2007, MSLR-10K, and MSLR-30K, with data statistics shown in Table 2. The dataset queries are partitioned into three subsets, namely training, validation, and test according to the 60%-20%-20% scheme. For each query-document pair of each dataset, relevance judgment $y$ is provided. The original feature sets of MSLR-10K/MSLR-30K have three behavior features (i.e., feature 134, feature 135, feature 136) collected from user behaviors. To reliably evaluate our method, we remove them in advance. MQ2007 only contains non-behavior features. Thus, at the beginning of our experiments, all datasets only contain non-behavioral features (i.e., $x^{nb}$). It is worth mentioning that there are other widely-used large-scale LTR datasets accessible to the public, such as Yahoo! Letor Dataset [9] and Istella Dataset [15]. However, they cannot be used in this paper because they contain behavior features but do not reveal their identity.

---

²https://anonymous.4open.science/r/EBRank-D039/

Table 2: Datasets statistics. For each dataset, the table below shows the number of queries, the average number of candidate docs for each query, the number of features, the relevance annotation $y$'s range, and the feature id of the BM25 to be used in our experiments.

| Datasets | # Queries | #AverDocs | # Features | $y$ | BM25 |
|---|---|---|---|---|---|
| MQ2007 | 1643 | 41 | 46 | $0-2$ | $25^{th}$ |
| MSLR-10k | 9835 | 122 | 133 | $0-4$ | $110^{th}$ |
| MSLR-30k | 30995 | 121 | 133 | $0-4$ | $110^{th}$ |

*5.1.2 Simulation of Search Sessions, Click and Cold-start.* Similar to prior research [7, 23, 46, 47], we create simulated user engagements to evaluate different LTR algorithms. The advantage of the simulation is that it allows us to do online experiments on a large scale while still being easy to reproduce by researchers without access to live ranking systems [33]. Specifically, at each time step, we randomly select a query from either the training, validation or test subset and generate a ranked list based on ranking algorithms. Following the methodology proposed by Chapelle et al. [10], we convert the relevance judgement to relevance probability according to $P(r = 1|d, q, \pi) = 0.1 + (1 - 0.1)\frac{2^y-1}{2^{y_{max}}-1}$, where $y_{max}$ is 2 or 4 depending on the dataset. Besides relevance probability, the examination probability $\rho_{rank(\pi,d)}$ are simulated with $\rho_{rank(\pi,d)} = \frac{1}{\log_2(rank(\pi,d)+1)}$. The $\rho_{rank(\pi,d)}$ is also the same examination probability used in Eq. 5 to compute $DCG$. For simplicity, we follow [33, 52, 54] to assume that users' examination $\rho_{rank(\pi,d)}$ is known in our experiment as many existing methods [3, 6, 51] have been proposed for it. With $P(r = 1|d, q, \pi)$ and $\rho_{rank(\pi,d)}$, according to Equation 2, clicks can sampled. We simulate clicks on the top 5 items, i.e., $k_s = 5$ in Equation 3. User clicks are simulated and collected for all three partitions. And we only use the sessions sampled from training partitions to train LTR models and sessions sampled from validation partitions to do validation. LTR models are evaluated and compared only based on test partitions. We collect clicks on validation and test queries in the simulation to construct the behavior features for their candidate document, which is used for inference only. In real-world LTR systems, behavior features are widely used in the inference of LTR models [9, 37].

The cold start scenario in ranking is an important part of our simulation experiments. We found two factors are essential for cold start simulation. Firstly, in real-world applications, new documents/items frequently come to the retrieval collection during the serving of LTR systems. To simulate new documents/items' coming, at the beginning of each experiment, we randomly sample only 5 to 10 documents for each query as the *initial candidate sets* $D_q$ and mask all other documents. Then, at each time step $t$, with probability $\eta$ ($\eta = 1.0$ by default), we randomly sample one masked document and add it to the candidate set $D_q$. Depending on the averaged number of document candidates and $\eta$, the number of sessions (time steps) we simulate for each dataset is

$$\#Session = \frac{\#Queries \times (\#AverDocs - 5)}{\eta} \quad (27)$$

where $\#Queries$ and $\#AverDocs$ are indicated in Table 2. The second factor for cold start simulation is that when a ranking algorithm usually is introduced in an LTR system, some documents/items already have collected user feedback in history. To simulate this,

for each query, we simulated 20 sessions according to BM25 scores to collect initial user behaviors for the initial candidate sets. The BM25 scores are already provided in the original datasets' features (see Table 2).

5.1.3   *Baselines.* To evaluate our proposed methods, we have the following ranking algorithms to compare,

**BM25**: The method to collect initial user behaviors.

**CFTopK**: Train a ranking model with Counterfactual loss that is widely used in existing works [30, 40, 54] and create ranked lists with items sorted by the model scores during ranking service.

**CFRandomK**: Train a ranking model the same way as CFTopK, but randomly create ranked lists with items during ranking service.

**CFEpsilon**: Train a ranking model the same way as CFTopK. Uniformly sample an exploration from $[0, 1]$ and add to each item's score from the model[54]. Create ranked lists with items sorted by the score after addition during ranking service.

**DBGD**[56]: A learning to rank method which samples one variation of a ranking model and compares them using online evaluation during ranking service.

**MGD**[42]: Similar to DBGD but sample multiple variations.

**PDGD**[31]: A learning to rank method which decides gradient via pairwise comparison during ranking service.

**PRIOR**[18]: When $x^b = 0$, the method will train a behavior feature prediction model to give a pseudo behavior feature $x^b$.

**UCBRank**[54]: uses relevance estimation from $x^{nb}$ when an item is not presented and uses relevance estimation from $x^b$ when an item has been presented. An upper Confidence Bound (UCB) exploration strategy is used as an exploration strategy.

**EBRank**: Our method shown in Algorithm 1.

For those baselines, DBGD, MGD, PDGD, CFTopK, CFRankdomK, CFEpsilon are unbiased learning to rank algorithms that try to learn unbiased LTR models with biased click signals. We will investigate, in this paper, whether they can overcome exploitation bias or not. We compare those methods with two feature settings. The first setting is that we only use non-behavior features $x^{nb}$, referred to as **W/o-Behav**. The second feature setting is that we use both behavior signals and non-behavior features. The feature setting is referred to as **W/-Behav(Concate)** when behavior-derived features and non-behavior features are concatenated together. **W/-Behav(Non-Concate)** means behavior signals and non-behavior features are combined using a non-concatenation way, such as the EB modeling used by EBRank. Method BM25 only has W/o-Behav results since it uses the non-behavior feature BM25 to rank items. CFTopK, CFRandomK, CFEpsilon, DBGD, MGD, and PDGD have ranking performance with both W/o-Behav and W/-Behav(Concate) feature settings, depending on whether behavior features are used or not. We follow Yang et al. [54] to use relevance's unbiased estimator $x^b = \frac{C}{n}$ as the behavior feature, with default value as 0 when $n = 0$. The default value for $x^b$ has a significant influence on methods DBGD, MGD, PDGD, CFTopK, CFRankdomK, and CFEpsilon since those methods will concatenate $x^b$ and $x^{nb}$. However, how to set the default $x^b$ for each method is not within the scope of this work, and we leave it for future works. PRIOR only has ranking performance with W/-Behav(Concate) because PRIOR is designed to relieve exploitation bias when behavior features and non-behavior features are concatenated. For UCBRank and EBRank, they only

have W/-Behav(Non-Concate) results since they have their own designed non-concatenation way to combine behavior signals and non-behavior features. However, our method EBRank is fundamentally different from UCBRank. UCBRank treats behavior features as independent evidence for relevance and linearly interpolates relevance estimation from $x^b$ and $x^{nb}$. Besides, UCBRank adopts an Upper Confidence Bound exploration strategy. However, EBRank interpolates relevance estimation from $x^b$ and $x^{nb}$ from a Bayesian perspective and uses the marginal-certainty exploration strategy derived in Eq. 16 to guide exploration.

5.1.4   *Implementation.* We retrain and update the prior model parameters periodically 20 times during the simulation. However, updating ranking logs, including user behaviors, is relatively easy and time-efficient, and we update them after each session. When we train the prior model according to loss in Eq. 13, we only train $\alpha = f_\theta(x^{nb})$ and fix $\beta = 5$ for simplicity, which works well across all experiments. Since DBGD and MGD are proposed to only work with linear models while other ranking algorithms do not have such limitations, we use linear models for the prior model of EBRank and all other baselines for fair comparison.

5.1.5   *Evaluation.* We evaluate ranking baselines with two standard ranking metrics on the test set. The first one is the Cumulative NDCG (**Cum-NDCG**) (Eq.7) with $\gamma = 0.995$ (same $\gamma$ used in [47, 48]) to evaluate the online performance of presented ranklists. The second metric is the standard **NDCG** (Eq.6), where each query's ranked list is generated by sorting scores from the final ranking models (excluding the exploration strategy part of each algorithm). The NDCG evaluates the offline performance of the learned ranking model. NDCG is tested in two situations: with (i.e., **Warm-NDCG**) or without (i.e., **Cold-NDCG**) behavior features collected from click history. In this way, our experiment can effectively evaluate LTR systems in scenarios where we encounter new queries with no user behavior history on any candidate documents (i.e., the *Cold-NDCG*) and the scenarios where user behavior exists (i.e., the *Warm-NDCG*). For simplicity, we set the rank cutoff to 5 and compute iDCG in both Cum-NDCG and NDCG with all documents in each dataset. Significant tests are conducted with the Fisher randomization test [44] with $p < 0.05$. All evaluations are based on five independent trials and reported on the test partition only.

## 5.2   Result

In this section, we will describe the results of our experiments.

5.2.1   *How does our method compare with baselines?* In Table 3, EBRank significantly outperforms all other methods and feature setting combinations on Warm-NDCG and Cum-NDCG, while its Cold-NDCG is among the best. The discussion in the following sections will give more insights into EBRank's supremacy.

5.2.2   *Will historical user behavior help ranking algorithms achieve better ranking quality?* As shown in Table 3, not all algorithms can benefit from incorporating behavior signals. Particularly, in Table 3, we indeed see that both W/-Behav(Concate) and W/-Behav(Non-Concate) feature settings help to boost most ranking algorithms to have better Warm-NDCG and Cum-NDCG. However, such boosting on Warm-NDCG and Cum-NDCG does not apply to all ranking algorithms, and there exist two exceptions. The first one is a trivial exception regarding CFRandomK. CFRandomK always randomly

**Table 3: The ranking performance. The best performances within each feature setting are bold. ∗ and † indicate statistical significance over other models in the same or all feature settings, respectively. Cold-NDCG and Warm-NDCG are the same within the W/o-Behav feature setting since behavior signals are not used in both settings.**

| Feature settings | Online Algorithms | MQ2007 | | | MSLR-10k | | | MSLR-30k | | |
|---|---|---|---|---|---|---|---|---|---|---|
| | | Cold-NDCG | Warm-NDCG | Cum-NDCG | Cold. | Warm. | Cum. | Cold. | Warm. | Cum. |
| | BM25 | 0.474 | 0.474 | 94.38 | 0.449 | 0.449 | 90.39 | 0.451 | 0.451 | 89.98 |
| | DBGD | 0.557 | 0.557 | 110.6 | 0.488 | 0.488 | 98.66 | 0.498 | 0.498 | 99.40 |
| | MGD | 0.562 | 0.562 | 110.9 | 0.473 | 0.473 | 95.72 | 0.502 | 0.502 | 101.3 |
| W/o-Behav | PDGD | **0.599** | **0.599** | 115.0 | **0.525** | **0.525** | **105.2** | **0.525** | **0.525** | **106.0** |
| | CFTopK | 0.591 | 0.591 | **116.7** | 0.510 | 0.510 | 102.9 | 0.506 | 0.506 | 101.6 |
| | CFRandomK | 0.589 | 0.589 | 87.69 | 0.509 | 0.509 | 79.59 | 0.514 | 0.514 | 78.63 |
| | CFEpsilon | 0.589 | 0.589 | 94.39 | 0.510 | 0.510 | 84.75 | 0.518 | 0.518 | 83.88 |
| | DBGD | 0.514 | 0.729 | 144.7 | 0.451 | 0.571 | 114.3 | 0.462 | 0.607 | 118.8 |
| | MGD | 0.523 | 0.725 | 142.1 | 0.461 | 0.558 | 109.0 | 0.444 | 0.595 | 122.5 |
| W/-Behav | PDGD | 0.574 | 0.745 | 147.6 | 0.466 | 0.591 | 117.9 | 0.480 | 0.584 | 116.4 |
| (Concate) | CFTopK | 0.385 | 0.579 | 113.5 | 0.369 | 0.489 | 97.53 | 0.366 | 0.491 | 97.65 |
| | CFRandomK | 0.377 | 0.771 | 87.11 | 0.403 | 0.596 | 79.82 | 0.404 | 0.603 | 78.91 |
| | CFEpsilon | 0.387 | 0.789 | 143.8 | 0.355 | 0.683 | 116.8 | 0.354 | 0.686 | 116.8 |
| | PRIOR | **0.597** | 0.791 | 158.7 | 0.507 | 0.554 | 110.3 | 0.503 | 0.557 | 111.4 |
| W/-Behav | UCBRank | 0.593 | 0.799 | 158.9 | **0.514** | 0.703 | 140.1 | 0.509 | 0.703 | 140.5 |
| (non-Concate) | EBRank(ours) | 0.596 | **0.849**∗† | **171.3**∗† | 0.513 | **0.762**∗† | **151.6**∗† | 0.513 | **0.762**∗† | **152.0**∗† |

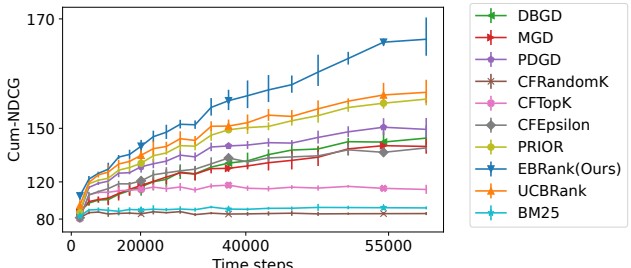

**Figure 3: Ranking performance (MQ2007, W/-Behav setting).**

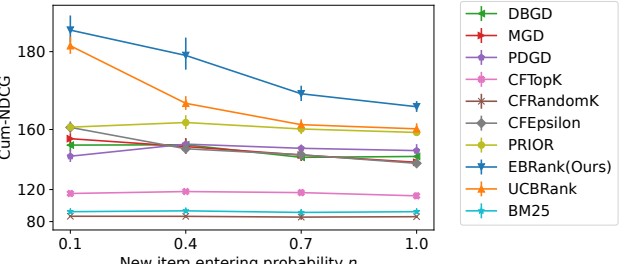

**Figure 4: Ranking performance with different entering probability $\eta$ in simulation (see Eq. 27) on MQ2007. We only consider using user behavior situations here (except BM25).**

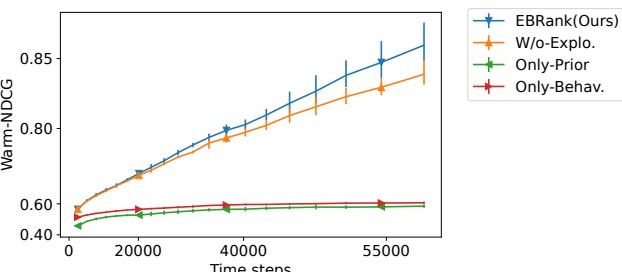

**Figure 5: Ablation Study of EBRank on dataset MQ2007. W/o-Explo. means excluding $MC(d)$ in Eq. 12 when generating ranklists. Only-Prior means only using the prior model part $\frac{\alpha}{\alpha+\beta}$ of $\hat{R}$ (in Eq. 10) to rank items. Only-Behav. means only using the behavior part $\frac{C}{n}$ of $\hat{R}$ (in Eq. 10) to rank items.**

ranks items and shows them to users, so its online performance, i.e., Cum-NDCG, can not be boosted. The second one is a **non-trivial exception** regarding CFTopK. Incorporating user behaviors even makes CFTopK degenerate on Warm-NDCG and Cum-NDCG for all three datasets. Besides CFTopK's degeneration on Warm-NDCG and Cum-NDCG, W/-Behav(Concate) feature setting even causes all ranking algorithms (except PRIOR) to experience a significant drop in Cold-NDCG, when compared to the W/o-Behav feature setting. Compared to other algorithms, UCBRank and our algorithm EBRank can benefit from behavior signals to have better Warm-NDCG and Cum-NDCG while avoiding drop in Cold-NDCG. In Table 3, PRIOR also avoids drop in Cold-NDCG but Prior is not as effective as UCBRank and EBRank in boosting Warm-NDCG and Cum-NDCG with user behavior. Besides Table 3, we additionally show ranking performance as time steps increase in Figure 3. As shown in Figure 3, EBRank consistently outperforms baselines.

*5.2.3 EBRank's robustness to entering probability*. To investigate the item entering speed ($\eta$)'s influence on the ranking performance, we show the experimental results with different $\eta$ in Figure 4. As shown in Figure 4, EBRank consistently significantly outperforms all other algorithms under different item entering speeds.

*5.2.4 Ablation Study*. In this section, we do an ablation study to see whether each part of our EBRank is needed. Due to limited space, we only provide analysis on MQ2007 dataset. As shown in

Figure 5, EBRank significantly outperforms the version only using the Behavior part or the prior model part to rank. Also, from the ablation study, we observe Bayesian modeling can help a ranking model reach good ranking quality and be robust to exploitation bias. The marginal-certainty-aware exploration additionally helps to discover relevant items, which helps to boost ranking performance in the long term.

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
