# OpenReview forum: "Mitigating Exploitation Bias in Learning to Rank with an Uncertainty-aware Empirical Bayes Approach"
_ACM.org/TheWebConf/2024/Conference — TheWebConf24_

### Official Review · Reviewer_W9vb · 2023-11-23

**Novelty:** 3
**Technical Quality:** 2

**Review:**

This paper targets the unbiased learning-to-rank problem with a particular focus on the exploitation bias. This paper proposes a new uncertainty-aware empirical Bayes-based ranking framework named EBRank. The experiments are conducted on three datasets (i.e., MQ2007, MSLR-10k, and MSLR-30k). The source code is provided via an anonymized repository.

While the proposed approach exhibits a certain degree of technical soundness and contributions, this paper requires substantial revision in terms of its presentation before being submitted for conference review.
- Importantly, this paper does not adhere to the provided submission template. For instance, (1) on the front page, the CCS concepts and paper information are deleted. (2) the margins between the sections have been excessively reduced, negatively impacting readability. (3) The tables lack polish and consistent styling, making them difficult to read.
- Many paragraphs consist of enumerated descriptions and lack proper organization, particularly in the experiment section. For instance, on page 7, a single paragraph occupies almost half of the page, making it difficult to discern the paper's intended message.
- There is no conclusion section in the paper.

**Questions:**

I don't have questions for this paper.

**Ethics Review Description:**

I don't have ethical concerns

**Reviewer Confidence:**

3: The reviewer is confident but not certain that the evaluation is correct

**Scope:**

4: The work is relevant to the Web and to the track, and is of broad interest to the community

---

### Official Review · Reviewer_FbaT · 2023-11-23

**Novelty:** 5
**Technical Quality:** 4

**Review:**

The paper introduces an uncertainty-aware empirical Bayes-based ranking algorithm called EBRank. EBRank uses a non-behavior feature-based prior model to get a prior relevance estimation. Specifically, EBRank uses the observed user behaviors to update posterior relevance estimation instead of concatenating behaviors as features in ranking models. Besides, EBRank additionally applies an uncertainty-aware exploration strategy to actively explore and collect user behaviors for empirical Bayesian modeling. Experiments on three public datasets show that EBRank outperforms state-of-the-art ranking algorithms.

The idea of proposing an uncertainty-aware algorithm for dealing with exploitation bias in learning to rank is interesting. The proposed approach is sound. The evaluation is conducted on three public benchmarks. The authors compare their method against nine state-of-the-art techniques. The evaluation methodology looks sound. Although the paper falls outside my main area of expertise, I think some aspects can be improved. In detail,

1) It is not clear why the authors employ gamma = 0.995 in the evaluation of the Cum-NDCG. It is clear it has been used in the past [47,48], but it looks like a magic number used for no specific reason. Please justify this point.

2) The statistical test used is the randomization test. No correction, e.g., Bonferroni, is applied to the reported statistical results. This means that the results reported cannot be statistically significant. Please refer to

https://sigir.org/wp-content/uploads/2018/01/p032.pdf
https://www.sigir.org/wp-content/uploads/2020/06/p14.pdf

3) The authors employ three datasets, but one (MSLR-10k) is a subset of another one (MSLR-30k). The results reported are always very similar between the two. The evaluation is also performed on a much smaller dataset, i.e., MQ2007. Why did authors not consider extending their evaluation to other datasets, e.g., the Istella datasets (https://istella.ai/data/letor-dataset/) or Yahoo! webscope? Please comment on that.

**Questions:**

1) Please comment on the use of correction. Are the results presented still statistically significant if correction is applied?
2) Please comment on the possibility of extending the datasets used to evaluate your proposed technique

The authors have answered the above questions during rebuttal.

**Ethics Review Description:**

not needed

**Reviewer Confidence:**

2: The reviewer is willing to defend the evaluation, but it is likely that the reviewer did not understand parts of the paper

**Scope:**

3: The work is somewhat relevant to the Web and to the track, and is of narrow interest to a sub-community

---

### Official Review · Reviewer_JpQJ · 2023-11-26

**Novelty:** 4
**Technical Quality:** 4

**Review:**

This paper addresses the challenge of exploitation bias in ranking systems. Exploitation bias refers to the potential discrimination against new items in a model overly reliant on behavioral features. The authors propose a novel ranking model, EBRank, based on Empirical Bayes. EBRank incorporates additional parameters, $\alpha$ and $\beta$, derived from the Beta distribution, enabling a new ranking strategy and loss function. Experimental results demonstrate that EBRank surpasses existing baselines in performance.


Advantages:

1. Exploitation bias is an good topic to be investigated.
2. The proposed method seems novel.
3.  The proposed method outperforms other baselines.



Disadvantages:

Disadvantages:

1. The paper's structure could be improved for better clarity. Section 4.1 introduces the methods but delays the necessary explanations, such as the derivation of Equation (11), which only appears at the end of Section 4. This can obscure understanding.

2. The paper lacks intuitive explanations for key equations, e.g.,  Equation (14) introduces a new objective but with just little explainations how it is related to exploitation bias problem.

3. For Equation (15), the reformulation of the estimated $DCG$ (5) to (15) is not clearly explained. The paper should give formal lemma in order to show that assumptions you made and whether they are strong or not, e.g., one assumption is the use of the Beta distribution.

**Questions:**

Please answer my questions mentioned above.

**Ethics Review Description:**

n.a.

**Reviewer Confidence:**

3: The reviewer is confident but not certain that the evaluation is correct

**Scope:**

4: The work is relevant to the Web and to the track, and is of broad interest to the community

---

### Official Review · Reviewer_Pne1 · 2023-11-29

**Novelty:** 5
**Technical Quality:** 5

**Review:**

The paper discusses the critical role of ranking in various artificial intelligence applications, emphasizing the prevalent use of learning-to-rank (LTR) models constructed from user behavior signals. The study challenges existing LTR algorithms that treat behavior and non-behavior signals indiscriminately, highlighting the potential suboptimal performance in practice. To address this issue, the authors introduce EBRank, an uncertainty-aware empirical Bayes-based ranking algorithm. EBRank utilizes a non-behavior feature-based prior model to estimate relevance, updating posterior relevance estimation dynamically using observed user behaviors. The algorithm also employs an uncertainty-aware exploration strategy to actively collect user behaviors. Experimental results on three public datasets demonstrate EBRank's effectiveness, practicality, and significant outperformance compared to state-of-the-art ranking algorithms.

The work is timely and relevant

Evaluation is rigorous using strong baselines

**Questions:**

questions to follow

**Ethics Review Description:**

no ethical issues

**Reviewer Confidence:**

2: The reviewer is willing to defend the evaluation, but it is likely that the reviewer did not understand parts of the paper

**Scope:**

4: The work is relevant to the Web and to the track, and is of broad interest to the community

---

### Official Review · Reviewer_zqXY · 2023-11-29

**Novelty:** 5
**Technical Quality:** 4

**Review:**

Summary:
The paper addresses the issue of exploitation bias in Learning to Rank (LTR) systems. These systems often rely on user behavior signals like clicks, which, while effective, can lead to bias since they are collected primarily on items already shown to users. The proposed solution is an uncertainty-aware empirical Bayesian-based ranking algorithm, EBRank, which differentiates between behavior and non-behavior signals in input features. EBRank employs a non-behavior feature-based prior model for initial relevance estimation and updates this estimation using observed user behaviors. It also incorporates an uncertainty-aware exploration strategy to balance exploration and exploitation. The effectiveness of EBRank is demonstrated through experiments on public datasets, showing significant performance improvements over state-of-the-art ranking algorithms.

Strengths:
1. The paper introduces a novel empirical Bayesian approach to address exploitation bias in LTR systems, a significant and often overlooked issue.
2. The experiments conducted on multiple public datasets offer a thorough evaluation of the algorithm's performance.
3. Addressing exploitation bias is highly relevant for improving the long-term performance of LTR systems in real-world applications.

Weaknesses:
1. The paper primarily focuses on clicks as user behavior data. Other types of user interactions, which could also influence ranking decisions, are not considered.
2. The proposed EBRank algorithm appears complex, and the paper lacks a detailed discussion on its scalability and computational efficiency in large-scale systems.
3.  While the algorithm shows improved performance on public datasets, it's unclear how it performs in more diverse real-world scenarios, especially with different types of user behavior and feedback mechanisms.

**Questions:**

1. How does EBRank handle new items or items with sparse user interaction data?
2. Is there an analysis of the computational complexity and scalability of EBRank in large-scale applications?
3. Could the authors elaborate on the potential limitations of the algorithm when applied to different types of user behaviors beyond clicks?

**Reviewer Confidence:**

3: The reviewer is confident but not certain that the evaluation is correct

**Scope:**

4: The work is relevant to the Web and to the track, and is of broad interest to the community

---

### Decision · Program_Chairs · 2024-01-22

**Decision:**

Accept

**Comment:**

This paper focuses on exploitation bias in learning to rank systems.

 The reviewers appreciated the importance of the problem and the experiments, but raised concerns with the focus on clicks, the complexity of the proposed model, the lack of diversity in the datasets used to experiment with the model (raised by two reviewers), and various issues with assumptions, justifications, structure, and clarity.

 The authors addressed most of these concerns in the rebuttal. The concerns could be addressed in the paper by integrating their rebuttal responses in the paper and doing some restructuring, adding explanations, and lemmas. There are also several assumptions made throughout the paper (e.g., gamma = 0.995) and these should be supported with sufficient justification and appropriate statistical testing (Bonferroni, etc.) where necessary. The authors should pay close attention to the remarks from reviewer W9vb on the presentation and structural issues.

 Overall, this paper was appreciated by reviewers and the issues that were raised appear within scope for an unshepherded revision for the conference. The reviewer giving the lowest scores mostly flagged presentation issues, which could/should be addressed for the next version. I am happy to recommend acceptance for the conference. Since there are no champions for this paper, I will select "Weak Accept" as my recommendation rating.